# Sex Lethal Gene Manipulates Gonadal Development of Medaka, *Oryzias latipes*, through Estrogenic Interventions

**DOI:** 10.3390/ijms232415496

**Published:** 2022-12-07

**Authors:** Tapas Chakraborty, Sipra Mohapatra, Michiya Matsuyama, Yoshitaka Nagahama, Kohei Ohta

**Affiliations:** 1Laboratory of Marine Biology, Faculty of Agriculture, Kyushu University, Fukuoka 819-0395, Japan; 2Aqua Bioresource Innovation Center (ABRIC), Faculty of Agriculture, Kyushu University, Fukuoka 819-0395, Japan; 3National Institute of Basic Biology, Okazaki 444-8585, Japan

**Keywords:** sex lethal, germ cell, mismigration, estrogen, estrogen receptor, medaka, gonad

## Abstract

Germ cells are pivotal for gonadal sexuality maintenance and reproduction. Sex lethal (*sxl*), the somatic sex determining gene of *Drosophila*, is the known regulator and initiator of germ cell femininity in invertebrates. However, the role of the Sxl homologue has rarely been investigated in vertebrates. So, we used medaka to clarify the role of *sxl* in vertebrate gonadogenesis and sexuality and identified two Sxl homologues, i.e., Sxl1a and Sxl1b. We found that *sxl1a* specifically expresses in the primordial germ cells (PGC), ovary, (early gonia and oocytes), while *sxl1b* distributions are ubiquitous. An mRNA overexpression of *sxl1a* accelerated germ cell numbers in 10 DAH XY fish, and *sxl1a* knockdown (KD), on the other hand, induced PGC mis-migration, aberrant PGC structuring and ultimately caused significant germ cell reduction in XX fish. Using an in vitro promoter analysis and in vivo steroid treatment, we found a strong link between *sxl1a* and estrogenic germ cell-population maintenance. Further, using *sxl1a*-KD and *erβ2*-knockout fish, we determined that *sxl1* acts through *erβ2* and controls PGC sexuality. Cumulatively, our study highlights the novel role of *sxl1a* in germ cell maintenance and sexual identity assignment and thus might become a steppingstone to understanding the commonalities of animal sexual development.

## 1. Introduction

In most species of higher eukaryotes, sexual reproduction is the favored form of metazoan reproduction and is critical for biodiversity and species management. During sexual reproduction, alleles from two different parents recombine together to increase variation and adaptation and eliminates the non-essential and deleterious mutation from a population. Species that reproduce sexually usually generate two types of individuals, males and females. Not surprisingly, the genetic programs that determine sex and control sexual differentiation need to be robust and should have various safeguards in place to ensure the survival of the population [1,2,3].

Germ and somatic cells are two essential variables in gonadal differentiation and sexual identity in vertebrates like mouse, chicken, and fish [4]. Among these, the primordial germ cell (PGC), a precursor of germ cell, stores genetic information for future generations and are the most important cells in the body. In vertebrates, PGCs arise at a distant site, divide, migrate through the gut mesentery and bloodstream, and arrive at and colonize at the gonadal primordium during the bipotential stage [4]. In addition, it has increasingly become clearer that, RNA-binding proteins (RBP) are also critical for germ cell sexuality and health. *rbp9* (*RNA-binding protein 9*), a *Drosophila* paralog of *elav* (*embryonic lethal abnormal vision*), is expressed during oogenesis and is required for female fertility [5]. *melav2* mRNA are expressed exclusively in the flat worm testis and play a crucial role in chromatin condensation and/or cell elongation and spermatid differentiation [6]. *zar1* (*zygotic arrest 1*) were also found to be important for zebrafish oogenesis, and the mutation of *zar1* caused significant upregulation of the zona pellucida proteins [7].

Sxl (Sex lethal), a member of the Sxl/Elav/Hur family RBP, is considered to be the major sex-determining factor in various invertebrates [8]. *sxl* directs the pre-mRNA splicing of *tra* (*transformer*) transcripts to produce the feminizing RNA-binding protein Tra-f and controls the sexual differentiation [9,10,11]. Briefly, *sxl* contains two RNA recognition motif (RRM), while the Elav and Hu protein family gene harbors three RRMs. It is considered that slight changes in RRM can lead to different interactions with other proteins, leading to various types of functions [12]. Deprivation of *sxl* activity in the somatic diplo-X cells of *Drosophila* results in lethality; however, the lack of *sxl* activity in the germline showed no apparent effect on germ cell viability [13], suggesting that *sxl* might play different role(s) in the germ cell. Transplantation of the *sxl* mutant pole cell (germ cell precursor in *Drosophila*) into a wild-type host female *Drosophila* caused excessive cystic proliferation of germ cells followed by restricted oogenesis and the occurrence of spermatocyte-like structures [14]. Later, it was found that a female biased post-transcriptional modification in the ovarian germ cell produced a longer and functional Sxl protein, which maintains its constitutive expression, probably through autoregulatory loop, and later accumulates in the oocyte and gets transported to fertilized eggs as a maternal source of *sxl* [15]. Recently, it has been determined that *sxl* acts to promote female identity in the germline by blocking reception of the JAK/STAT signal in male germline stem cells, a precursor for male and female gametes [16]. In this regard, it was found that *sxl* transiently expresses in XX PGC during PGC migration and helps them to gain female identity in *Drosophila* [17]. Transient overexpression of *sxl* in XY PGC drives oogenesis, suggesting their significance in germ cell sexuality maintenance [17]. Recently, some of the alternatively spliced Sxl homologues from Pacific white shrimp have been reported to express in germ cells and regulate gonadal development and sexual maintenance [18].

In medaka (*Oryzias latipes*), an important model species for gonadal sex determination and differentiation studies [19], the early gonadal settlement of PGCs is regulated by Sdf1/Cxcr4-mediated chemotactic migration [20,21]. Later, during medaka sex differentiation, both germ and somatic cells co-operatively regulate the gonadal development in an estrogen-dependent manner [21] and help the PGCs to undergo a proliferative mitosis and meiosis in females while restricting proliferation in males. Estrogen is known to directly affect the transcriptional profiles of several major sex-related genes (e.g., *dmrt1*, *gsdf*, *aromatase*, and *rspo1*), germ cell proliferation characteristics, and sexual identity in medaka [reviewed in 2]. Earlier, we found that, during early sex differentiation, *erb2* predominantly expresses in the germ cells (PGC) of embryonic medaka, and the alternation of *erb2* influences the germ cell settlement in gonad and thereby affecting the gonadal sexuality. However, the mechanisms involving PGC protection is largely unknown. In a recent study, it was shown that an RNA-binding protein like *Ol-cug-bp1 (Oryzias latipes CUG-binding protein)* and *Ol-bsf (Oryzias latipes Bicoid Stability Factor)* antagonistically regulates *dmrt1bY*/*dmy,* the sex-determining gene of medaka, through a conserved 30 bp 3’UTR sequence [22,23] and controls its stability in the PGC. This highlights the importance of RNA-binding proteins in gonadal sex management. However, there is a wide knowledge gap, which, if cleared, will allow us to understand the commonalities and differences of sexual development across taxas. So, in this study, we have isolated two different sxl1 homologues and analyzed their tissue distribution and cellular localization and evaluated the effects of steroid, overexpression, and knockdown to understand the role of RNA-binding proteins, like *sxl*, in the development and maintenance of gonadal sexuality in medaka.

## 2. Results

### 2.1. Cloning and Phylogenetic Analysis

Using an advanced distant homology search, we have located two Sxl sequences in medaka chromosomes 21 and 22 and isolated the full-length cDNA from the medaka embryonic cDNA library using specific primers. We found that medaka *sxl1a* contain 1038 bp ORF, encoding 345 amino acids, while medaka *sxl1b* consists of 1622 bp ORF, encoding 356 amino acids. Further analysis suggested that Sxl1a harbored one complete and one partial RRM, while Sxl1b possessed two complete RRMs. Further protein structure analysis highlighted that both have very close structural similarity with various Sxl and Elavl proteins (Figure 1A–C). Phylogenetic analysis suggested that Sxl1a and sxl1a2 are close relatives of *Drosophila* Sxlf, while the Sxl1b is distantly related to any other Sxl or Elav family gene (Figure 1C), thus enhancing the possibility that Sxl might play a significant role in medaka and vertebrates.

### 2.2. Tissue Distribution, Ontogeny and Cellular Localization

Several Sxl- and Hur-like proteins were identified from the medaka embryonic cDNA library. Interestingly, *sxl1a* was strongly expressed in the ovary, while *sxl1b* showed ubiquitous expression (Figure 2A). Ontogeny analysis during embryonic stages showed that both *sxl1a* and *sxl1b* start a zygotic expression as early as Stage-15 (Mid gastrula stage) (Figure 2C,D). However, only *sxl1a*, but not *sxl1b*, showed a female-biased expression throughout embryonic and adult stages. On the other hand, *dazl* (a germ cell marker) showed a sex-biased expression from S-33 (Figure 2B). *sxl1a* mRNA were found to be localized specifically in the embryonic PGC (at 2daf) and the early gonia and oocytes of the adult gonad (Figure 3A,C). An *ISH* (*in situ* hybridization) analysis failed to detect any *sxl1a* mRNA in the adult testis (Figure 3E). Contrastingly, a *WISH* (Wholemount *ISH*) analysis depicted a strong *sxl1b* expression in optic regions which later became ubiquitous (Figure 3B). *sxl1b* expressions were also seen in various testicular cells and oocytes of the adult gonad (Figure 3D,F). This suggests that *sxl1a* and *sxl1b* might have some role in sex-biased germ cell/gonad development and maintenance. Of note, due to practical limitations in raising specific antibodies against sxl1a and sxl1b, protein analysis was not conducted.

### 2.3. Functional Validation of sxl Action in Gonad Development

To find out the role of *sxl1a* action, we micro-injected Sxla5′utr-sxlaorf-GFP-3′utrsxl mRNA into one-cell-stage embryos, evaluated the GFP localization (Figure 4A), and found that *sxl1b* spreads throughout various cell types (Figure 4B) while *sxl1a* particularly localizes in the germ cell (Figure 4C), further confirming the *ISH* data. Interestingly, when checked at 10 DAH, the *sxl1a* overexpressed XY fish had increased germ cell population (Figure 4D). Changes in germ cell number is a classic indication of gonadal sex change in medaka, and *sxl1* is widely known to regulate germ cell health in various invertebrates. Additionally, careful observations of embryos hybridized with *sxl1a* and *olvas* with the *ISH* probe and the overexpression analysis data highlighted that all *vasa*-positive cells are loaded with *sxl1a*, although some *olvas* negative cells, which housed ample amounts of *sxl1a* mRNA, were also existent. So, to pinpoint the *sxl* role in the medaka gonad, we knockdowned (KD) the *sxl1a* using a previously established protocol [24]. We found a strong decline in *sxl1a* expression at S-36 (Figure 4E), while no significant difference in *sxl1b* was recorded in the *sxl1a*-KD fish (Figure 4F). As expected, the gonads of both XX and XY had a smaller number of germ cells at 10 DAH (Figure 4D). However, when the gonadal development was evaluated, especially the PGC migration and settlement, we found that PGCs of *sxl1a*-KD embryos were disoriented from as early as 2 daf (Figure 5A). Interestingly, contrast to their control counterparts, some of the PGCs in *sxl1a*-KD, after reaching the gonad showed structural abnormality. The mismigrated and abnormal germ cells specifically expressed high amount of autophagic cell death marker (LC3) proteins at 4 daf. Later we manually counted the total germ cell (TGC), mismigrated germ cells (MGC) and abnormal germ cells (AGC) using the Z-stake confocal images of whole embryo (n = 9) and found that at both 2 and 4 daf, especially in the *sxl1a*-KD embryos, there was significant increase in MGC and AGC as compared to their control counterpart (Figure 5A,B). On the other hand, *sxl1b* overexpression did not affect the gonadal development while *sxl1b* knockdown caused embryonic lethality (Table 1). These observations suggest that *sxl1b* is critical for embryonic development while *sxl1a* might have acquired a new function in gonadal maintenance of medaka.

### 2.4. Effect of Sex Steroids

Sex steroids are an integral part of medaka sex-biased gonad development. Numerous reports suggest that exposure to exogenous estrogen and androgen can alter germ cell characteristics and subsequently drive the XY-male to XY-female and XX-female to XX-male sex reversal, respectively. To understand the sex steroid-*sxl1* relation, we treated the embryos and adults with 17β-estradiol (E2), 11-ketotestosterone (11-KT), and aromatase inhibitor (AI) for different spans of time (maximum 10 days). As expected, *sxl1a* were decreased in 11-KT treated XX embryos and adults, while a low but significant increase was observed in E2-treated XY groups (Figure 6A). On the other hand, 11-KT-treated fish showed an increase in *sxl1b* expression during adult stages (Figure 6A). Surprisingly, after 10 days of AI treatment, the *sxl1a* transcription elevated exponentially **(**Figure 6A). To confirm these results, we performed a separate experiment for a period of 21 days and collected samples daily and stored half for transcriptional analysis and the other half for germ cell counting, using previously published protocol [25]. The freshly collected gonadal parts were dissociated using Accumax (Stemcell Technologies, Tokyo, Japan), filtered through 20 μm mesh, fixed with PFA, stained with vasa antibody (GeneTex, Irvine, CA, USA), and counted using a cell analyzer (EC800, Sony). We found a gradual increase in vasa-positive cells till 12 DAT (days after treatment) and a reduction thereafter (Figure 6B). We also observed that AI treatment gradually increases the *sxl1a* expression till 12 DAT and falls thereafter. Interestingly, *gsdf1* (a male-biased somatic cell marker) expression starts increasing from 12 DAT (Figure 6B). A strong positive correlation (0.88) between germ cell number and *sxl1a* expression was also noted. This might imply that *sxl1a*, and maybe not *sxl1b*, acts in an estrogen-dependent manner to control germ cell sexual identity.

### 2.5. Effect of Steroid Receptors

Notably, we have reported that the reduction of *erβ2* causes germ cell mis-migration, an excessive accumulation of autophagic proteins, a reduction in germ cell number in the development, and hence female to male sex reversal, further insinuating that *erβ2* and *sxl1a* might cross paths in germ cells. Hereafter, we have focused our attention to understand the role of *sxl1a* in the gonad maintenance of medaka.

Fueling our hypothesis, an in silico analysis of the *sxl1a* promoter depicted several ERE binding sites. So, to confirm any direct interaction between *sxl1a* and *er* (estrogen receptor), we performed the dual luciferase assay, using various lengths of the promoter and found that at least 4 kb of the promoter could significantly affect the luciferase activity in a dose-dependent manner. Notably, one half-ERE (AGGTCA) was detected in −4 kb promoter region of *sxl1a*. Further analysis depicted that E2 (Figure 6C,D) and other estrogenic chemicals could also increase the luciferase activity (data not shown). Specifically, the luciferase activity was also significantly higher in *erβ2* (Figure 6C) than *erα* (Figure 6D). To further validate the *sxl1a*-*erβ2* interconnection, we checked the *erβ2* cellular localization in *sxl1a*-KD fish and found that they were congregated in germ cells (Figure 5A), as seen in the case of the Sxl1a-GFP expression in Sxl1a-OV fish (Figure 4B). Later, we also measured the *sxl1a* expression in *erβ2*-KO fish and *erβ2* expression in *sxl1a*-KD fish in various stages of the medaka embryo. Earlier, we found that, in the normal medaka embryo, the germ cell-specific *erβ2* expression increases from S-26 to S-36 and the somatic cell-specific *erβ2* expression dramatically decreases from S-33 [26]. So, in this investigation we specifically included these stages to confirm the involvement of *erβ2* in *sxl1a* transcriptional regulation. Interestingly, the *sxl1a* expression did not change until S-26; however, from S-33 they showed a significant decline in *erβ2*-KO fish (Figure 7A). To our surprise, we did not find any significant decrease in the *erβ2* expression in *sxl1a*-KD fish till S-36 (Figure 7B). These data suggest that *erβ2* might be instrumental in managing germ cell-specific *sxl1a* transcription, and the reduction of *sxl1a* aggravates other sex-biased pathways to initiate a chain reaction of sex change. Notably, it is well documented that *gsdf1* antagonistically regulates *erβ2* action [26] and the *tra* gene is a long-known associate of *sxl* [9]. So, we measured the transcription of *gsdf1* and *tra2* in *sxl1a*-KD and *erβ2*-KO fish and found that the *gsdf1* expression significantly increased in *sxl1a*-KD-XX fish from S-33 (Figure 7C), while the *tra2* expression significantly downregulated in the *sxl1a*-KD-XX and *erβ2*-KO-XX fish from S-26 and S-36, respectively (Figure 7D). Strikingly, when the *sxl1a* expression in normal female medaka during early stages were correlated with the respective *erβ2* expression, a strong correlation (Table 2) was observed. This cumulatively indicates that *sxl1a* strongly interacts with *erβ2* and regulates germ cell development, and any discrepancy in their regulation could ultimately affect germ cell health and thereby sexual transdifferentiation.

## 3. Discussion

Germ cell development is pivotal in reproduction and species management. Several genes, diverged due to genome duplication, have been identified to have a functional role in germ cell regulation. In this work, for the first time in fish, we have identified two different Sxl1 isoforms from medaka, one specifically expressing in the female germ cells and the other having ubiquitous expression. Further, overexpression and knockdown analysis confirmed the distinct role of *sxl1a* and *sxl1b* in germ cell maintenance and embryonic development, respectively. A promoter analysis suggested a steroid/steroid receptor responsive regulation of *sxl* isoforms, and further analysis confirmed the role of *sxl1a* in the regulation of *erβ2*-dependent germ cell maintenance.

It is well accepted that, during evolution, the genomes of most vertebrates had two rounds of whole genome duplications [27], whereas the genomes of most bony fish had another extra round (3-round duplication), resulting in the evolution of the modern fish [28,29,30]. An in silico genome analysis revealed that the two different types of RRMs containing proteins (one having two RRMs and the other having three) are present in separate chromosomes of medaka. Elav proteins, harboring three RRMs, are common in both invertebrates and vertebrates. However, proteins containing two RRMs, especially Sxl, have only been characterized from invertebrates [9,10,11] and are known to have functions in invertebrate sexual development. Our phylogenetic analysis suggests that Sxla and Sxlb are distinctly grouped together with invertebrate Sxl and vertebrate Elav/Elavl clade, suggesting the existence of the *sxl* gene in vertebrates, at least in medaka. In a different study [18], it was found that, even among invertebrates, Sxl1 shows a strong divergence between crustacea and insects. This, along with the fact that Sxl1a and Sxl1b contain different RRMs, further hints that Sxl has evolved sporadically during evolution and has a strong notion of neofunctionalization. Notably, in various occasions, it has been determined that changes in RRM protein structure helps the Elav family genes to attain diverse functions [12]. Though Elav family genes are mostly linked to embryonic development, the ovary specific Elav-like transcripts, functional even after zygote formation, were detected in the ovarian follicular cells of a *Xenopus* frog [31] and thus highlighted the importance of Elavl (embryonic lethal abnormal vision-like protein) proteins in germ cell formation [31]. In our phylogenetic analysis, we also found that the medaka Sxl1a is closely related to both invertebrate Sxl and vertebrate Elavl protein, further suggesting their possible roles in gonadal physiology.

*sxl*, in addition to being a sex determining gene in *Drosophila melanogaster*, also acts as a somatic cell identity creator [32,33]. In XX animals, *sxl* becomes activated and imposes female development, while in XY animals, *sxl* remains inactive and thus male development prevails [8]. We found a strong expression of *sxla* in the ovary but not in the testis of medaka, and a sexually unbiased expression of *sxl1b* in both ovary and testis. However, *sxl* in a housefly, *Musca domestica*, did not show such sexual variation, and, when ectopically expressed in *Drosophila*, did not affect the known targets of endogenous *sxl* and caused embryonic lethality in both sexes [34]. This indicates that the importance of *sxl* in sex determination is restricted to certain species, although they maintain sex differentiation roles in various species.

The function of the Sxl/Elav/Hur family proteins can vary depending on the cellular localization. Since Elav/Hur RBPs can shuttle between the nucleus and cytoplasm, they also most likely exert gene-specific functions depending on their cellular localization [35]. In our research, we found a strong localization of *sxl1a* in the germ cell (both nucleus and cytoplasm) and germline stem cell precursors, while *sxl1b* expressed only in the somatic and germ cells. Similarly, only few of the alternatively spliced isoforms of *sxl* were explicitly expressed in the germ cell and oocytes of Pacific white shrimp gonads, pointing towards the fact that *sxl* is vital for gonads and germ cells. Further, it was found that transient, but natural, expressions of *sxl* in the female PGCs of *Drosophila* are critical for female-identity maintenance [17]. It was found that *HuB* is maternally provided in the zebrafish embryo and exhibits a germ cell-specific expression during embryogenesis. However, restriction of the *HuB* mRNA to the germ cells is dependent on several sequence elements in its 3′UTR, which acts to degrade the mRNA in the soma and stabilizes it in the germ cells by interacting with *dazl* [22]. When these UTR sequences were deleted, the same mRNA are distributed ubiquitously. Similar 3′utr sequence was observed in the 3′utr of *sxl1a*, but not on *sxl1b*, and we observed a germ cell-specific expression of *sxl1a* and a ubiquitous distribution of *sxl1b*. This implies that, probably due to these conserved 3′utr sequences, *sxl1a* was able to specifically retain in the germ cells and absence of such elements allowed a ubiquitous expression of *sxl1b*. Additionally, it was demonstrated that Ol-bsf, a stabilizer of *dmrt1* in the germ cell, are abundant in the germline stem cells during adulthood. Notably, RNA-binding proteins are known to regulate poly-A shortening in mammalian cells [36]. The presence of RNA-binding proteins, like *Ol-bsf* [23] and *sxl1a*, in the germline stem cells might protect these stem cells from early poly-A shortening and maintain pluripotency. In this regard, we found that, during the secondary sex change in medaka, these germline stem cells serve as a precursor for sexual transdifferentiation (unpublished data), and it would be very interesting to study the role of *sxl* in stem cell transdifferentiation.

Gonadal sexuality is regulated by the balancing actions of sex-biased genes, sex steroids and various other physiological tugs of war. Among them, estrogen and its receptors are essential for germ cell development, maintenance, gonadal sexuality, and sexual transdifferentiation [reviewed in 2]. The presence of ERE (Estrogen Recognition Element) is a potential indicator of an estrogen-related action of a candidate gene. Due to the presence of ERE, it is quite possible that the *sxl1* gene is regulated by estrogenic action in medaka. Further, it was found that *sxl*, when specifically overexpressed in male *Drosophila* PGC, was enough to induce male to female transdifferentiation. Additionally, we found that 11KT and AI reduces the *sxl* expression in females and E2 increases its expression in males. Similar alternations of the *erβ2* expression have been reported earlier [26], and we too found a strong correlation of *erβ2* with *sxl1a*. Reports have suggested that the *erβ2* expression remains female biased but maintains a minimum level in males. In the present work, the corresponding *sxl1a* expression was also recorded in males, and it is highly likely that, due to extra estrogen in the system, *erβ2* shifts its gears and promotes *sxl1a* transcription. The *sxl1a* mRNA then gets protected by *erβ2* in the germ cell and paves the way for further development. Supporting this observation, the addition of AI reduced the aromatase activity, causing reduction of estrogen availability and subdued *erβ2* action and further suppression of the *sxl1a* expression. Such estrogenic actions are very quick, but our data shows that the *sxl1a* expression continued to increase till 10 DAT and reduced thereafter, suggesting a simultaneous feedback loop. Temporally speaking, male-biased genes like *gsdf* were also upregulated after *sxl1a* decreased in the AI-treated gonad, indicating that the body requires a certain period to prepare itself for any transdifferentiation activity. In this regard, the *sxl1a* expression reduces after a lag in *erβ2*-KO fish. All together, these suggest that *sxl1a* might have some important part in gonad maintenance and combining the possible contribution of *sxl1a* in stem cell maintenance, it is likely that *sxl1a* has some executive role in sexual plasticity.

Notably, the RNA-binding protein *tra* and its cofactor *tra2* are active participants in females, whereas *sxl*, *tra*, and *tra2* are assumed to be non-functional in males [37]. In the present investigation, the knockdown of *sxl1a* reduced the *tra2* expression in the female gonad thus evincing conserveness of the *sxl-tra2* actions in fish. In a recent study, Olcug-bp1, an upstream regulator of *dmrt1* stabilization in gonad, was found to alter the *tra2* expression [23]. In the same study, it was found that an overexpression of *Ol-bsf* correlates with an increased stabilization of the D3U-box–containing mRNAs, while an overexpression of Ol-cug-bp1 correlates with a decreased stabilization of the D3U-box–containing mRNAs in the gonad, altering the sexual development [23]. This revelation opens newer prospects to study the estrogen-dependent mechanism of sxl regulation in medaka more intimately.

Estrogen regulates the RBP action in various organisms through estrogen receptors. For instance, E2 treatment enhances the *clep5* expression in a rat’s hippocampus and thereby controls *erβ* splicing and its action on brain aging [38]. In another study, it has been demonstrated that zebrafish *zar1*, an RNA-binding protein regulating RNA translation, mutants show oogenesis arrest and female-to-male sex reversal [39,40]. Earlier, we found that *erβ2* specifically expresses in the early germ cell, and a reduction of *erβ2* causes a mis-migration of germ cells and germ cell death and thereby sex reversal [26]. In the present experiment, we found a strong localization of both *erβ2* and *sxl1a* in the germ cell, thus conveying the fact that the *sxl1a* and *erβ2* actions are interlinked. Of note, the *erβ2* expression initiates around S-18, while *sxl1a* starts its zygotic expression from S-12, a bit earlier and most importantly, in *erβ2*-KO fish, the *sxl1* expression eventually subsides, but with a lag. Moreover, in *sxl*-KD fish, the *erβ2* expression does not take a fall till 0 DAH. The significant reduction in the *erβ2* expression after that might be linked to the *sxl*-KD-associated germ cell death. This germ cell reduction, a classic signal for a sex change in medaka, probably initiates a chain of reactions, especially the local germ-soma interaction, causing an increase in *gsdf* and other male-biased gene expressions and tilting the balance towards gonadal masculinity [26]. But, based on the fact that *erβ2* and *sxl1a* co-localize together, it is highly likely that *erβ2* regulates the *sxl1a* post-translational stability in medaka, similar to *hur* mRNA stability through *dazl* in zebrafish. Additionally, the RNA-binding protein *hur* plays a critical role in stabilizing *erα* mRNA [40], suggesting that at least some RNA-binding proteins, including medaka *sxl1a*, are regulated through an estrogen/estrogen receptor pathway to regulate germ cell health. In *Drosophila*, *ssx* (sister of sex lethal) controls the auto-regulatory splicing of *sxl* by competing for the RNA-binding protein sites in *sxl1* and safeguards the male sexuality, which otherwise in the absence of *ssx*, as in females, allows the *sxl* to form an autoregulatory loop [41]. It is evident from our study that *erβ2* is a coregulator of *sxl1a* in medaka, and it is quite possible that *sxl1a*-*erβ2* co-ordinated actions help the female to maintain the *sxl* level above the threshold necessary for femininity maintenance. It is also likely that other Elav proteins or genes like *Ol-bsf* safeguard the accidental overexpression of *sxl1a* in males. However, in depth analyses are pertinent to comprehending the *sxl*-based germ cell sex reversal mechanisms.

In summary, we found that a female-biased expression of *sxl1a* and not *sxl1b* is essential for germ cell sexuality and femininity maintenance in medaka. We also determined that *sxl1a* functionality is regulated by an estrogen-estrogen receptor-mediated positive feedback loop and controls germ cell health. Future investigation in the direction of *sxl*-mediated gonadal stem cell maintenance might shed light on vertebrate gonadal differentiation, sexual transdifferentiation, sexual plasticity, and overall sex management.

## 4. Materials and Methods

### 4.1. Plasmid Construction

pGEM-T-easy plasmid of different genes were used for in situ hybridization (*ISH*) probe synthesis and real time PCR standard preparation, whenever necessary. Expression plasmids were constructed using pcDNA3.1(+) vector backbone, using complete ORFs of required genes, if not specifically mentioned. Various lengths of putative promoter regions of *sxl1a* were isolated from medaka genome using specific primer set and cloned into pGL3 basic vector (Promega, Tokyo, Japan) using HD-Infusion-cloning kit (Takara Bio, Shiga, Japan), as mentioned earlier [26]. The overexpression and knockdown plasmids were prepared using previously published protocol [24,26]. Briefly, Sxl1a–5-utr, sxl1a ORF-GFP and Sxl1a 3-utr, Sxl1b2–5utr, and Sxl1b ORF-GFP and Sxl1b 3-utr were sequentially cloned into pCS2 vector and named as pCS2-sxl1a-GFP and pCS2-sxl1b-GFP, respectively. The relevant antisense fragment of sxl1a was PCR amplified, cloned into pcDNA3.1(+) vector in an antisense orientation and named as pcDNA-Sxl1aAS. Plasmid DNA, used for downstream experiments, was purified using the plasmid purification kit (Qiagen, Tokyo, Japan).

### 4.2. Experimental Animals

The cab strain of medaka, *Oryzias latipes*, was used for this study, if not specifically mentioned. Fish were maintained at 26  ±  2 °C under a photoperiod comprising 14-h lightness and 10-h darkness. Eggs were collected within 30 min of fertilization and incubated in distilled water (Milli-Q) containing an antifungal solution (methylene blue, 2–3 ppm) at 26  ±  2 °C. Brooders and juveniles were fed with fresh artemia, while larvae were given artificial food. All experimental protocols herein used were approved by the Institutional Animal Care and Use Committee of Kyushu University.

### 4.3. Sample Collection

Samplings were carried out at stage (S) 8, 12, 15, 18, 22, 25, 27, 33, 36, 39 and 0 days after hatching (DAH) following the reported description about medaka development [42]. At each stage, 192 embryos were individually sampled and stored in DNA/RNA shield and only after confirming the genetic sex [19], 6 individuals/sex were pulled together to form an experimental sample. A total of 10 such experimental samples were prepared at each stage. From 10 DAH, only the experimental gonads were similarly pooled to form analysis group.

### 4.4. Phylogenetics and Protein Analysis

Protein and cDNA sequences were obtained from the NCBI, (http://www.ncbi.nlm.nih.gov/, accessed on 1 June 2022), with accession numbers XP_024122996.1 (*Oryzias melastigma*, Elavl1a); XP_019953635.1 (*Paralichthys olivaceus*, Elavl1); XP_042245133.1 (*Thunnus maccoyii*, Elavl1a); XP_030248070.1 (*Sparus aurata*, Elavl1a); XP_026196260.1 (*Anabas testudineus*, Elavl1a); XP_022606306.1 (*Seriola dumerili*, Elavl1a); XP_012713704.1 (*Fundulus heteroclitus*, Elavl1a); NP_001069922.1 (*Bos taurus*, Elavl1); NP_001410.2 (*Homo sapiens*, Elavl1); NP_001102318.1 (*Rattus norvegicus*, Elavl1); XM_008289964.1 (*Stegastes partitus*, Elavl2), (*Tetraodon rubipres*, Elavl1); AAB41913.1 (*Homo sapiens*, Hur, Human antigen R); BT009793.1 (*Homo sapiens*, Elav); AAB17967.1 (*Mus musculus*, Elavg); AGI44577.1 (*Macrobrachium nipponense*, Sxl1); QCZ24930.1 (*Cherax quadricarinatus*, Sxl1); XP_046438003.1 (*Daphnia pulex*, Sxl1); XP_032778830.2 (*Daphnia magna*, Sxl); ROT79577.1 (*Penaeus vannamei*, Sxl); XP_004085828.1 (*Drosophila melanogaster*, Sxl); NM_167118.2 (*Drosophila melanogaster*, Sxlb); NM_080052.4 (*Drosophila melanogaster*, Sxlc); NM_167117.2 (*Drosophila melanogaster*, Sxlf); NM_001031893.2 (*Drosophila melanogaster*, Sxlm); XM_047011654.1 (*Drosophila willistioni*, Sxl); XM_002056740.3 (*Drosophila viridis*, Sxl) and used for sequence analysis. The putative amino acid sequences were translated using expasy translation tools (http://web.expasy.org/translate, (accessed on 1 June 2022), Expasy 3.0, SIB, Lausanne, Switzerland). The putative signal peptides and domains were analyzed using InterProScan 5 (http://www.ebi.ac.uk/Tools/pfa/iprscan5, (accessed on 1 June 2022), EMBL, Cambridgeshire, UK). The secondary structural analysis, i.e., helix formation, helix–helix interactions, and beta and gamma turns were performed using ProFunc (http://www.ebi.ac.uk/thornton-srv/databases/profunc, (accessed on 1 June 2022), EMBL, Cambridgeshire, UK). Further, the nuclear localization signal and the in silico protein 3D structures were predicted using Predict protein (www.predictprotein.org, (accessed on 1 June 2022), Rostlab, Garching, Germany) and SWISS MODELLING (http://swissmodel.expasy.org, (accessed on 1 June 2022), SIB, Lausanne, Switzerland), respectively.

### 4.5. Tissue Distribution Analysis

Total RNA was isolated from the ovary, testis, brain, muscle, heart, liver, kidney, and intestine of 8-month-old adult fish with RNeasy kit (Qiagen, Germantown, MD, USA) following the manufacturer’s directions. cDNA was synthesized with 1 µg of the total RNA using superscript IV (Invitrogen, Tokyo, Japan). Gene-specific primers were employed for the RT-PCR analysis. Positive and negative controls were set up to validate the distribution pattern with plasmid DNA and water as templates, respectively. *ef1α* was amplified (as an internal control) from medaka to test the quality of the cDNAs used in the PCR [43,44]. The PCR conditions consisted of initial denaturation at 94 °C (3 min), followed by 25 cycles of 94 °C (30 s), 58 °C (30 s), and 72 °C (2 min and 30 s). The reaction was ended by a further incubation for 10 min at 72 °C. PCR was performed on a GeneAmp PCR system (9700 thermal cycler, Applied Biosystems, Foster City, CA, USA) and products were separated on 1.2% agarose gels.

### 4.6. Quantification of Changes in Gene Expression by Realtime PCR

Changes in gene expression were quantified using the Light cycler 480-II sequence detection system (Roche, Tokyo, Japan). Total RNA was isolated from embryos or gonads using Direct-zol-96 MagBead DNA-RNA kit (Zymo Research, Irvine, CA, USA). The genomic DNA were used for fish sex determination. cDNA synthesis was carried out using superscript IV (Invitrogen, Waltham, MA, USA) from 100 ng of total RNA. The first strand cDNAs were diluted to 100 μL for subsequent use. Gene-specific real time PCR was performed using SsoFast EvaGreen Supermix (Biorad, Tokyo, Japan) and 5 μL of cDNA, according to the manufacturer’s instructions. The PCR conditions included an initial denaturation at 94 °C (2 min) followed by 40 cycles at 94 °C (30 s) and 60 °C (1 min). *ef1α*, *βactin*, and *rps18* were used as internal controls. The absolute transcript copy number of each gene was determined with the help of appropriate standard curves and normalized with the average of *ef1α*, *βactin*, and *rps18* copy numbers in each sample [26]. The reported values are averaged from experimental triplicates, if not otherwise mentioned. The specificity of primer sets, throughout this range of detection, was confirmed by the observation of a single amplification product of the expected size, melting curve, Tm (melting temperature), and sequences. All assays were quantified, with standard curves (mean Ct vs. log cDNA dilution) having slopes between −2.99 and −3.34, a linear correlation (R^2^) between the mean Ct and the logarithm of cDNA dilution of >0.985 in each case. All test cDNAs were run in triplicates for each gene. The primers were designed according to unmutated (wild) DNA sequences, if not otherwise mentioned.

### 4.7. Histology In Situ Hybridization (ISH) and Immunohistochemistry

An amount of 4% paraformaldehyde (PFA)-fixed paraffin-embedded samples (at least 10 fish per group) were used for *ISH*. Bouin-fixed 10 DAH gonads were similarly processed, stained with Hematoxylene and Eosin (HE), and used for germ cell counting, using predetermined protocol [26,43]. All the histological analyses were performed using 5 µm sections. Whole mount *ISH* (*WISH*) was conducted using 4% PFA-fixed dechorionated embryos, using previously published protocol [45]. Multicolor *WISH*-WIHC (whole mount immunohistochemistry) were conducted using established protocols [26,44], wherever necessary. For *ISH*, probes for sense and anti-sense digoxigenin-labelled RNA strands were transcribed in vitro with RNA-labelling kit (Roche Diagnostics GmbH, Mannheim, Germany) from plasmid DNA, containing 450–500 bp ORF fragments of respective genes. Sections were deparaffinized, hydrated, treated with proteinase K at 10 µg/mL (Roche), and hybridized with the sense or anti-sense DIG-labelled RNA probe at 60 °C for 18–24 h. The hybridization signals were then detected using an alkaline phosphatase-conjugated anti-DIG antibody (Roche) and NBT, as described previously [43]. Multicolor *WISH* probes were similarly prepared using FISH Tag™ RNA Multicolor Kit (Thermofisher Scientific, Tokyo, Japan). Agarose embedded multicolor *WISH* samples were subjected to confocal microscopy (LSM 700, Zeiss, Jena, Germany).

### 4.8. Promoter Analysis

The genomic sequence of *sxl1a* was taken from Medaka Ensemble database (http://www.ensembl.org/Oryzias_latipes/Info/Index, accessed on 10 November 2021) and analyzed using Promoter 2 (http://www.cbs.dtu.dk/services/Promoter/, accessed on 5 January 2022). The estrogen responsive elements (ERE) were manually marked according to their respective conserved sequences [46]. 4–6 kb promoter of *sxl1a* was isolated from medaka genomic DNA and directionally cloned in pGL3 promoter-less luciferase vector. The promoter analysis was performed using previously described protocol [26,43]. Briefly, HEK-293 cells were seeded in 24-well plates at 5 × 10^5^ cells/well in Dulbecco’s modified Eagle’s Medium (Sigma-Aldrich, Tokyo, Japan). After 24 h, the cells were transfected with pGL3-promoter-luciferase plasmid and pcDNA3.1-ERβ2/ERα plasmid (200 ng/well) in triplicates. The luciferase assay was performed after 48 h of transfection. The experiment was repeated thrice for reproducibility.

### 4.9. Overexpression and Knockdown of sxl Genes

Both knockdown and overexpression analysis were performed using dsred-olvas 3′utr transgenic medaka. Knockdown (KD) and overexpression (OV) were performed using the previously published protocols [24,26]. Briefly, the mRNAs were synthesized from pCS2-sxl1a-GFP and pCS2-sxl1b-GFP plasmids with mMESSAGE mMACHINE SP6 kit (Ambion, Austin, TX, USA) following polyA addition with poly-A tailing kit (Ambion). The purified RNA (in case of OV experiment) or linearized pcDNA-Sxl1aAS plasmid (in case of KD experiment) were injected in one-cell-stage medaka embryos (olvas-dsred transgenic) at 1 ng/µL. Pictures were taken at 2 and 3 days after fertilization (DAF) using Zeiss LSM 700 confocal microscope for each embryo, separately. The Z-stage confocal sections of each live embryo were further analyzed to ascertain the PGC numbers at each stage. The embryos showing reporter gene (GFP) expression were then grown separately until 10 DAH on a 24-well dish (1 embryo/well) and fixed/preserved for histological and real time PCR analysis.

### 4.10. Chemical Treatment

The chemicals, 17β-estradiol (E2, Sigma, Ronkonkoma, NY, USA), aromatase inhibitor (AI, LKT labs, St Paul, MN, USA), and 11-keto testosterone (11-KT, Sigma, USA) were dissolved in DMSO (Nacalai Tesque, Kyoto, Japan) and used for treatment. For embryonic treatment, one-/two-cell-stage medaka embryos (100 embryos/50 mL of water) were treated till 18 DAF (if not otherwise specifically mentioned). The embryos were reared as mentioned above, with daily water exchange. The adults (15/sex/group) were treated for 10–21 days with daily water exchange. The experiments were repeated at least 5 times and each individual was processed separately.

### 4.11. Data Analysis

All experiments were conducted for a minimum of 6 times (biological replicates) and statistical differences were assessed based on biological replicates, if not otherwise mentioned. Statistical differences in relative mRNA expression between various experimental groups were assessed by one- or two-way ANOVA of normalized data, followed by Tukey’s test, or Student’s *t*-test. All statistical analyses were performed using SPSS, version 22, and all experimental data are shown as mean ± SEM. Differences were considered statistically significant at *p* < 0.05, if not otherwise mentioned. The correlations were calculated using Pearson correlation coefficient method.

## Figures and Tables

**Figure 1 ijms-23-15496-f001:**
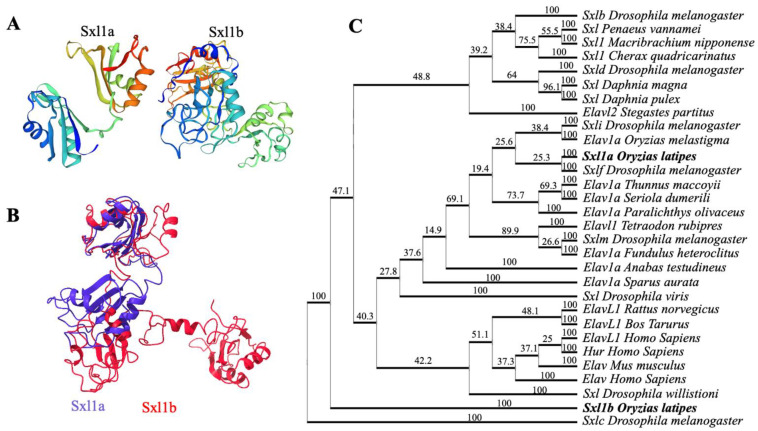
**Putative protein structure and phylogenetic analysis of two Sxl1 isoforms in medaka.** Three-dimensional modeling was performed to visualize the monomeric structure of Sxl1a and Sxl1b (**A**) proteins, and both structures were superimposed using Chimera-X 1.5 software to depict the similarities between medaka Sxl1 homologues (**B**). Phylogenetic analysis (**C**) was carried out using ClustalW. The tree was generated with PHYLIP and viewed with the Phylodendron online software. Values on the tree represent bootstrap scores of 1000 trails. *Drosophila melanogaster* Sxlc was used as an out-group.

**Figure 2 ijms-23-15496-f002:**
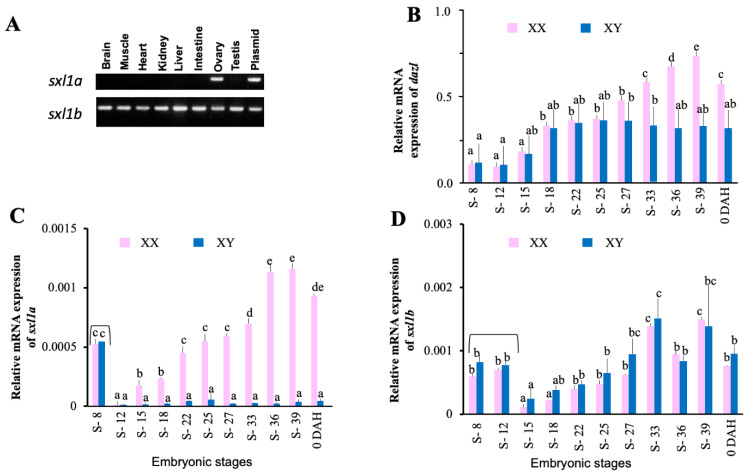
**Tissue distribution and mRNA expression analysis of *sxl* homologue in medaka.** Real time PCR analysis of the two isoforms of *sxl1* in various tissues of adult medaka (**A**), where B, brain; M, muscle; H, heart; K, kidney; L, liver; I, intestine; O, ovary; T, testis; P, plasmid. Changes in mRNA levels of *dazl* (germ cell-related gene) (**B**), *sxl1a* (**C**), and *sxl1b* (**D**) in XX and XY embryos of medaka collected at stage (S) 8, 12, 15, 18, 22, 25, 27, 33, 36, 39, and 0 DAH. Note: Data are shown as the mean ±  SEM and expressed as relative abundance and corrected for the average of *ef1α*, *βactin*, and *rps18;* different letters denote significant differences at *p* < 0.05. Black Bracket-Maternal expression.

**Figure 3 ijms-23-15496-f003:**
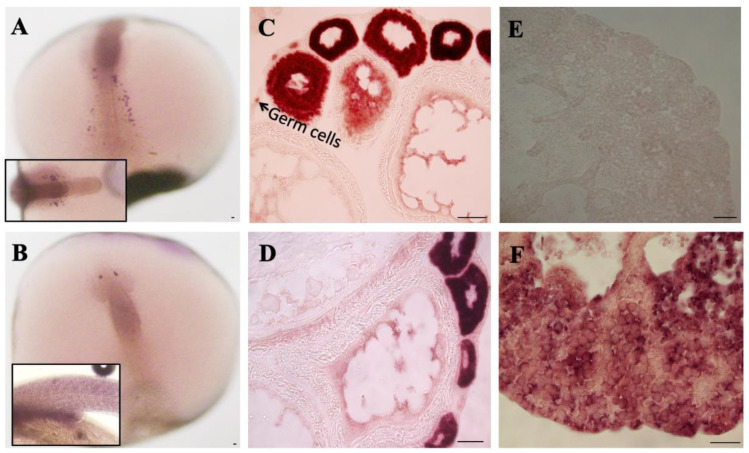
**Cellular localization of *sxl1a* and *sxl1b* in embryos and adults by *WISH* and *ISH*.** At stage (S)-17, *sxl1a* was specifically expressed in germ cell precursors (**A**), while *sxl1b* was localized in the optic region (**B**). Insets in both (**A**) and (**B**), marked with a black boundary, show the *olvas* expression at S-17 and *sxl1b* expression at S-33, respectively. Interestingly, during adulthood, *sxl1a* was specifically expressed in the female germ cell and early oocytes (**C**) but not in testis (**E**). On the other hand, *sxl1b* was abundant in the female early oocytes (**D**) and both germ and somatic cells of testis (**F**). Bar length: 50 µm.

**Figure 4 ijms-23-15496-f004:**
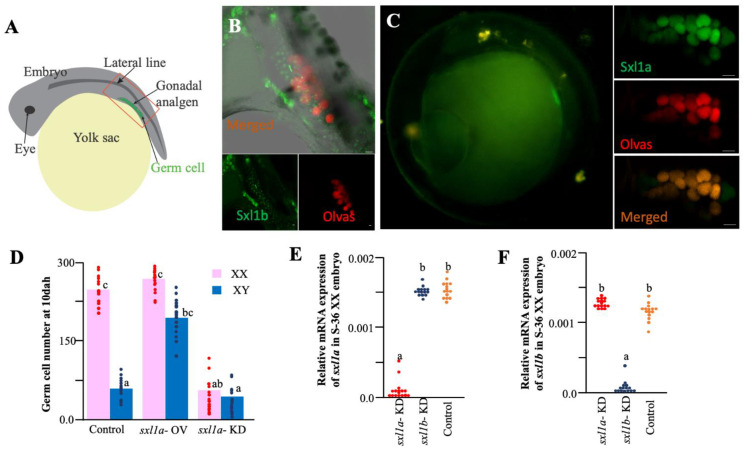
**Functional analysis of *sxl1a* and *sxl1b* in medaka embryo.** We evaluated the GFP localization in gonadal analgen and its nearby regions (“Region of Interest” marked with red box) of overexpressed embryos (**A**). When overexpressed, Sxl1b-GFP were concentrated in various embryonic body parts at 3daf (**B**), while Sxl1a-GFP were specifically congregated in the PGCs of Olvas-dsRED transgenic medaka at 3 DAF (**C**). 15 overexpressed and knockdown embryos were raised till 10 DAH, and germ cells were counted using serial sections of bouin fixed samples (**D**). At 10 DAH, the total germ cells were significantly increased and dramatically reduced in XY-*sxl1a*-OV and XX-*sxl1a*-KD embryos, respectively (**D**). Average germ cell numbers at each stage were plotted as columns, and each dot represents one individual. Further, changes in *sxl1a* (**E**) and *sxl1b* (**F**) mRNA abundance were measured using *sxl1a*-KD, *sxl1b*-KD, and control embryos at S-36 (15 XX embryos/group) and plotted onto graphs. Each dot represents one individual. Note: Different letters denote significant differences at *p* < 0.05; pictures are general representations from 15 individual observations. Bar length: 10 µm.

**Figure 5 ijms-23-15496-f005:**
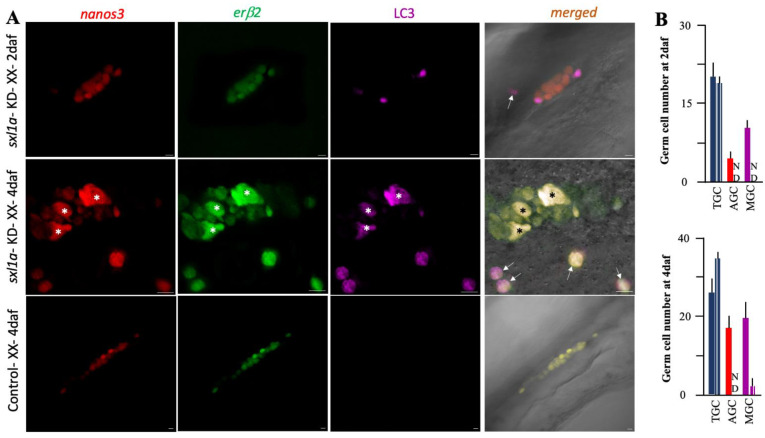
**Evaluation of germ cell physiology in *sxl1a*-KD medaka embryos.** The 2 and 4 DAF *sxl1a*-KD (upper and middle panel, respectively), and control (lower panel) embryos were fixed, and multicolor *ISH*-IHC was performed using a *nanos3* and *erβ2 ISH* probe and LC3 antibody and analyzed using Z-stack confocal imaging (n = 9/group). Representative photomicrographs portray germ cell characteristics in various injected groups (**A**). Mis-migrated germ cells are marked with an arrowhead, and abnormal autophagic germ cells are marked with an asterisk (*). Each type of cells from nine individuals/groups were manually counted and plotted into graphs (**B**). Note: Data are shown as the mean  ±  SEM and expressed as relative abundance, corrected for the average of *ef1α*, *βactin*, and *rps18*. Blue, red, and pink column, respectively, represent the total germ cell (TGC), abnormal germ cell (autophagic and distorted cell structure, AGC) and mis-migrated germ cell (MGC). ND—Not detected. Solid and stripped columns represent KD and control embryos, respectively. Bar length: 10 µm.

**Figure 6 ijms-23-15496-f006:**
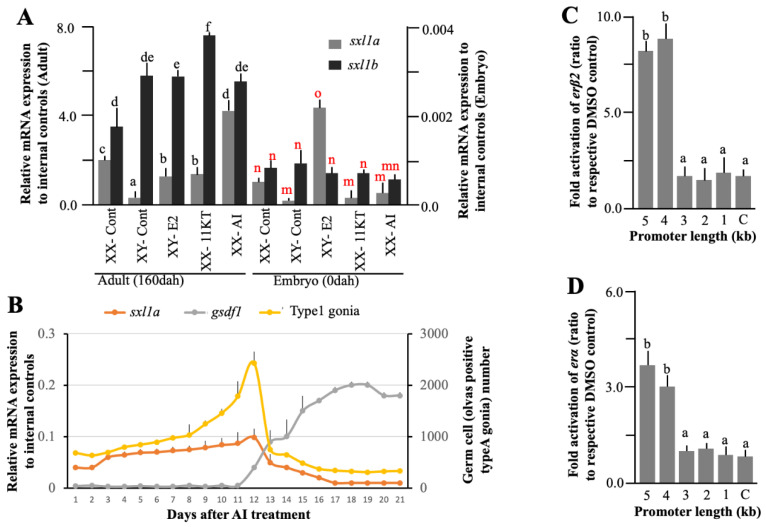
**Effects of sex steroids on *sxl1* regulation in medaka.** In vivo analysis, using E2 (1ng/mL), 11-KT (10ng/mL), and AI (25ng/mL) depicted a strong and significant alteration of *sxl1a* but relatively mild changes of *sxl1b* in both adult gonad and embryos (experimental replicate = 4). Steroid concentrations were adjusted using pilot experiment (**A**). Further in-depth analysis using AI-treated mature adult XX female demonstrated a strong positive correlation (0.88) between *sxl1a* and germ cell population (**B**). *gsdf1* expression was assessed to measure the masculinization of the female. Data are shown as the mean  ±  SEM and expressed as relative abundance corrected for the average of *ef1α*, *βactin*, and *rps18*; different letters denote significant differences at *p* < 0.05. In vitro promoter analysis under the influence of *erβ2* (**C**) and *erα* (**D**) of *sxl1a* showed estrogen/*ER*-dependent modulations. The luciferase activity of E2-treated samples were normalized with their respective DMSO control, and the mean relative fold activations were plotted in the graph. C—promoter-less control; experimental replication—6; different letters denote significant differences at *p* < 0.05.

**Figure 7 ijms-23-15496-f007:**
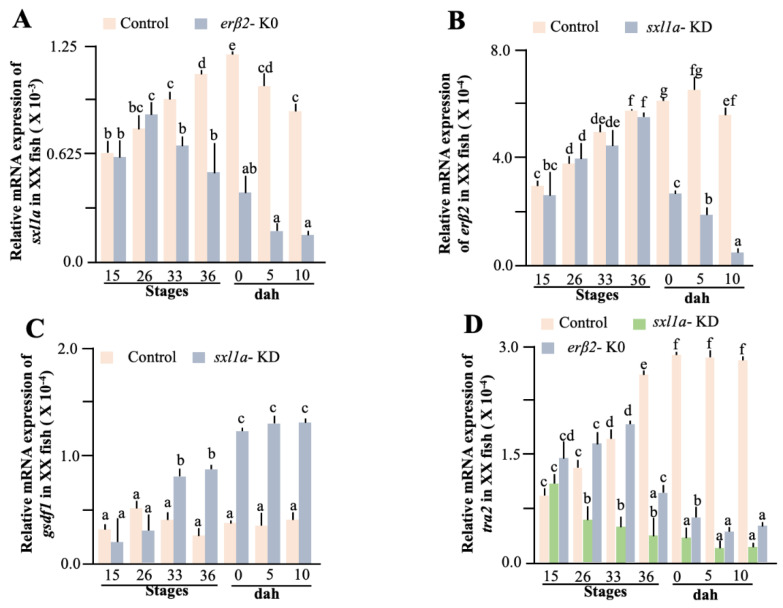
**Evaluation of *sxl1a* and *er**β2* interaction in XX female medaka.** Transcriptional alterations of *sxl1a* (**A**)*, erβ2* (**B**), *gsdf1* (**C**), and *tra2* (**D**) were measured using various embryonic stage samples of control-XX, *sxl1*-KD, and *erβ2*-KO medaka. Data are shown as the mean  ±  SEM and expressed as relative abundance, corrected for the average of *ef1α*, *βactin*, and *rps18*; different letters denote significant differences at *p* < 0.05.

**Table 1 ijms-23-15496-t001:** Survivability evaluation of various microinjected (% death) medaka embryos till hatching.

Group (Total Injected Embryos)	1daf	2daf	4daf	6daf	Hatching	Total
Control (151)	5.30 ±1.1	2.65 ± 1.2	2.65 ± 1.1	0.66 ± 0.4	1.99 ± 1.1	13.25 ± 0.9
*sxl1a*-OV (164)	4.27 ± 1.4	3.05 ± 1.1	2.44 ± 1.1	1.83 ± 1.0	1.22 ± 0.5	12.80 ± 0.8
*sxl1b*-OV (173)	5.20 ± 1.3	2.89 ± 1.4	2.31 ± 0.4	2.31 ± 0.8	1.16 ± 0.3	13.87 ± 0.6
*sxl1a*-KD (171)	4.68 ± 1.4	2.92 ± 1.2	3.51 ± 1.5	2.34 ± 1.0	0.00 ± 0.0	13.45 ± 1.0
*sxl1b*-KD (175)	75.43 ± 12.4	18.86 ± 6.5	5.71 ± 4.8	0.00 ± 0.0	0.00 ± 0.0	100.00 ± 0.0

Note: Eggs were collected from 20 parental pairs and randomly selected for injection for each of the five groups. Injections were carried out in six batches; each batch corresponds to the injected groups from day 1. Controls were injected with the mixture of linearized PCS2 and pCDNA 3.1 vector. OV—overexpression.

**Table 2 ijms-23-15496-t002:** Analysis of correlation of *sxl1a* with candidate male and female-biased genes of medaka during embryonic development.

Gene	Correlation with *sxl1a*
*er* *β2*	0.91
*foxl2*	0.34
*foxl3*	0.39
*cyp19a1*	0.21
*olvas*	0.84
*gsdf1*	−0.87
*dmrt1*	−0.27
*sox9a2*	−0.63

Note: Relative transcription (normalized against average of *ef1a*, *βactin* and *rps18*) of each candidate gene at each embryonic stage were used for the correlation analysis. All embryonic stages (S-12, 15, 18, 22, 25, 27, 33, 36, 39 and 0 DAH) were used for this analysis.

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
