# Peer review of "Sex Lethal Gene Manipulates Gonadal Development of Medaka, Oryzias latipes, through Estrogenic Interventions"

_ijms, 2022, doi:10.3390/ijms232415496_

Round 1

Reviewer 1 Report

Sex lethal (sxl) is the somatic sex-determining gene that controls sexual development by regulating alternative splicing. In this manuscript, the authors investigated the expression and roles of Sxl homolog in medaka organisms. The authors identified two Sxl isoforms, Sxl1a and Sxl1b, which differentially expresses in medaka tissues. The authors further investigate the specific roles of two Sxl isoforms and identify a unique role of Sxl1a in germ cell maintenance and sexual identification. This study is novel and contributes to the study of sex determination in various organisms. I have a few comments to help this manuscript improve.

1. Figure 1 A-B, it’s unclear how to interpret the similarities between Sxl1a and Sxl1b from 3D modeling. One suggestion is to try a superposition of these two structures and analyze the similarities.

2. Figure 1C, how to interpret the distance is vague.

3. Figure 2B, how did the authors separate and collect embryos in different stages?

4. Throughout Figure 3, there are no bars

5. On lane 131, what is ISH analysis? The explanation should be here for the first it shows.

6. Figure 4, Sxl1b and Sxl1a are over-expressed and examined for their localization, which might not represent the endogenous proteins, in addition, sxl1b and sxl1a are tagged with GFP which might affect their localization.

7. Figure 4A-B, it’s very difficult to understand the spatial distribution of different cell types all through the panels, the author could use a graphic figure and label the spatial distribution of cells to assist the IF images. Besides, are the ROI used in panels A and B the same?

8. Throughout the figures, either bars or bar lengths are missing.

Author Response

  1. Figure 1 A-B, it’s unclear how to interpret the similarities between Sxl1a and Sxl1b from 3D modeling. One suggestion is to try a superposition of these two structures and analyze the similarities.

Answer: We have added the superimposed structure. It seems that, though they matches with other sxl and ELAV protein but Sxl1a and Slx1b are slightly different from each other. Based on the data, we have modified the relevant sections.

  1. Figure 1C, how to interpret the distance is vague.

Answer: We apologize for the inconveniences. We have added the distances in the figure 1C.

  1. Figure 2B, how did the authors separate and collect embryos in different stages?

Answer: each individual embryos were stored in DNA/RNA shield and subsequently used for DNA and RNA isolation. After confirmation of genetic sex of each individual using PCR (Matsuda et al., 2002), the required number of samples from each sex were pulled together to form an analytical sample for cDNA preparation. We have added it in section 4.3.

  1. Throughout Figure 3, there are no bars

Answer: Corrected

  1. On lane 131, what is ISH analysis? The explanation should be here for the first it shows.

Answer: We appreciate reviewer’s suggestion. We have corrected the mistake.

  1. Figure 4, Sxl1b and Sxl1a are over-expressed and examined for their localization, which might not represent the endogenous proteins, in addition, sxl1b and sxl1a are tagged with GFP which might affect their localization.

Answer: We thank the reviewer for his concern. We understand that, overexpression and GFP tagging might affect the actual localization. However, our WISH analysis showed a high resemblance with the overexpression data. Moreover, we overexpressed the 5`UTR- sxl1-gfp-3` UTR mRNA to ascertain the protein localization. Additionally, 5` UTR- GFP-3`UTR constructs of each sxl1 gene were preliminarily tested, also depicted same localization results (data not shown). So, at least for this case we are quite certain that overexpression experiments resembles endogenous expression.

  1. Figure 4A-B, it’s very difficult to understand the spatial distribution of different cell types all through the panels, the author could use a graphic figure and label the spatial distribution of cells to assist the IF images. Besides, are the ROI used in panels A and B the same?

Answer: We are thankful for reviewer’s suggestion. We have added a general graphical figure showing the germ cell, gonad and area of interest. In both case the ROI were same. For simplicity we marked the ROI in the graphical image.

  1. Throughout the figures, either bars or bar lengths are missing.

Answer: We have corrected the errors.

Reviewer 2 Report

The paper entitled as "Sex lethal gene manipulates gonadal development of medaka, Oryzias latipes, through estrogenic interventions" is very interesting. However, I have certain suggestions that I want authors to incorporate while revising manuscript:

1. In the introduction authors should elaborate how sexual reproduction cleanse harmful mutation from a population.

2. Authors should correct the spelling of zona pellucida

3. Why focus so much on Drosophila sex determining pathway in the introduction section. Be precise

4. In the Result A section- Why authors did not perform western blot to check sxla and sxlb expression

5. In Figure 5- What is the no. of sample size

6. Knockdown was done ubiquitously or tissue specific knockdown was performed. What is the percentage of knockdown in both the cases

7. Why authors did not measure the expression of sxl a in sxl b KD fishes and vice versa

8. erB2 receptor is only localised to gonads or also expressed in brain. Justify with the experiment

Author Response

The paper entitled as "Sex lethal gene manipulates gonadal development of medaka, Oryzias latipes, through estrogenic interventions" is very interesting. However, I have certain suggestions that I want authors to incorporate while revising manuscript:

  1. In the introduction authors should elaborate how sexual reproduction cleanse harmful mutation from a population.

Answer: Thank you for your suggestion. It was reported that, sexual process can serve as a mechanism to “filter out” aberrations at the chromosome level. Moreover, during sexual reproduction any harmful mutations were diluted or discarded depending on whether they were associated with the sex-controlling gene. We have modified the portion accordingly.

  1. Authors should correct the spelling of zona pellucida

Answer: Incorporated.

  1. Why focus so much on Drosophila sex determining pathway in the introduction section. Be precise

Answer: We have modified the introduction as suggested.

  1. In the Result A section- Why authors did not perform western blot to check sxlaandsxlb expression

Answer: We appreciate the reviewer’s concern. We agree with the fact that, it is good to have western blot data. However, we failed to generate a good pair of antibodies for medaka. We will surely continue trying and include in our future research.

  1. In Figure 5- What is the no. of sample size

Answer: 9 samples from each group were used for analysis.

  1. Knockdown was done ubiquitously or tissue specific knockdown was performed. What is the percentage of knockdown in both the cases

Answer: Knockdown was performed ubiquitously. We obtained ~100% knockdown effect in both cases, at least in respective gene expression. The data are in included in figure 4E and 4F.

  1. Why authors did not measure the expression of sxl ainsxl b KD fishes and vice versa

Answer: we have added the data as Figure 4E and 4F.

  1. erB2 receptor is only localised to gonads or also expressed in brain. Justify with the experiment

Answer: Thank you for your concern. ERb2 receptor largely localizes in the gonad during embryogenesis. However, previously, we published that erb2 starts expressing in the brain also from 2 days after fertilization (Chakraborty et al., 2019) and affects brain type aromatase expression. As sxl1a is localized in the gonad, in this experiment we only focused on the gonadal expression.